# Electron pairing and nematicity in LaAlO₃/SrTiO₃ nanostructures

Aditi Nethwewala [1,2], Hyungwoo Lee[3], Jianan Li[1,2], Megan Briggeman[1,2], Yun-Yi Pai[1,2], Kitae Eom [3], Chang-Beom Eom[3], Patrick Irvin[1,2] & Jeremy Levy [1,2,4] ✉

Strongly correlated electronic systems exhibit a wealth of unconventional behavior stemming from strong electron-electron interactions. The LaAlO₃/SrTiO₃ (LAO/STO) heterostructure supports rich and varied low-temperature transport characteristics including low-density superconductivity, and electron pairing without superconductivity for which the microscopic origins is still not understood. LAO/STO also exhibits inexplicable signatures of electronic nematicity via nonlinear and anomalous Hall effects. Nanoscale control over the conductivity of the LAO/STO interface enables mesoscopic experiments that can probe these effects and address their microscopic origins. Here we report a direct correlation between electron pairing without superconductivity, anomalous Hall effect and electronic nematicity in quasi-1D ballistic nanoscale LAO/STO Hall crosses. The characteristic magnetic field at which the Hall coefficient changes directly coincides with the depairing of non-superconducting pairs showing a strong correlation between the two distinct phenomena. Angle-dependent Hall measurements further reveal an onset of electronic nematicity that again coincides with the electron pairing transition, unveiling a rotational symmetry breaking due to the transition from paired to unpaired phases at the interface. The results presented here highlights the influence of preformed electron pairs on the transport properties of LAO/STO and provide evidence of the elusive pairing "glue" that gives rise to electron pairing in SrTiO₃-based systems.

Many remarkable properties of electronic materials can be traced to the presence of strong electron-electron interactions and their coupling with other degrees of freedom. Unconventional superconductivity, various forms of magnetism, and electronic nematicity are some notable examples in this domain. The term nematicity was first used in the context to liquid crystals to describe the system's crystalline symmetry breaking from C4 to C2. Over time the definition of nematicity has evolved to other domains including the electronic nematic phases based on various theoretical models[1]. Electronic nematicity is characterized by the rotational symmetry breaking of an electronic fluid due to anisotropic electron interactions, resulting in strongly anisotropic transport behavior which can be tunable with chemical potential or chemical doping, and also by a magnetic field[2]. Electronic nematic phases have been found to exist in a wide range of electronic materials[1] extending from GaAs/AlGaAs heterostructures[3,4] to high-temperature superconductors[5–7] and twisted bilayer graphene[8]. Theoretical frameworks developed to help understand the origin of electronic nematicity face challenges because of the wide

[1]Department of Physics and Astronomy, University of Pittsburgh, Pittsburgh, PA 15260, USA. [2]Pittsburgh Quantum Institute, Pittsburgh, PA 15260, USA. [3]Department of Materials Science and Engineering, University of Wisconsin-Madison, Madison, WI 53706, USA. [4]Present address: Department of Physics and Astronomy, University of Pittsburgh, Pittsburgh, PA 15260, USA. ✉e-mail: jlevy@pitt.edu

range of systems that exhibit this behavior[1]. In strongly correlated systems such as high-temperature superconductors, electronic nematicity is often observed in the pseudogap regime[9]. The precise connection between electronic nematicity and pseudogap behavior has empirical support but is not well established[5,10–12].

Strontium titanate (STO) is the first and best-known superconducting semiconductor[13]. STO-based heterostructures, and in particular formed with LaAlO$_3$ (LAO)[14] inherits the superconducting properties from STO and exhibit two-dimensional (2D) superconductivity without the need for chemical doping[15]. Prominent features include a characteristic dome shape of the superconducting critical temperature[16,17], evidence for a pseudogap phase up to $T \sim 500\mathrm{mK}$[15], and evidence for electron pairing without superconductivity, seen in single-electron transistors (SETs)[18], and within quasi-1D straight ballistic nanowires[19–23]. The characteristic magnetic field, $B_p$ at which electrons unbind can be two orders of magnitude larger than the boundary for superconductivity ($B_p \gg B_c \sim 0.2\mathrm{T}$) and the paired electron states are stable at temperatures as high as 900 millikelvin (mK), well above the superconducting transition temperature ($T_c \sim 300\mathrm{mK}$) where the Pauli limit does not apply[24]. Hence, the paired-but-non-superconducting regime covers a significant region of parameter space, overlapping in temperature and magnetic field with a wide range of experiments performed on macroscopic devices[25]. However, the influence of non-superconducting paired states on the transport properties has not been explored.

One class of macroscopic transport experiments involves anisotropic magnetoresistance (AMR) which has been explored by several groups[26–29]. Here we summarize representative results by Joshua et al.[26]. These experiments are performed in a Hall bar geometry in which an in-plane magnetic field $H^{\parallel}$ is applied. Below a critical magnetic field, $H_c^{\parallel}$[26,29] the anisotropy of the magnetoresistance is solely determined by the direction of magnetic field. However, above $H_c^{\parallel}$, an additional component of anisotropy appears with the pinning of AMR along preferred directions[26]. In the same parameter regime of carrier density and magnetic field, the onset of an anomalous Hall effect (AHE) is also observed[26]. Hall measurements show a change in the slope of the Hall resistance, at the critical magnetic field, $H_c^{\parallel}$. The magnitude of $H_c^{\parallel}$ depends sensitively on the carrier density of the system[26]. Both AMR and AHE have been linked to a Lifshitz transition[17] in which electrons from the $d_{xz}$ and $d_{yz}$ bands appear in addition to the lower energy $d_{xy}$ band. However, multiband theory cannot fully account for AMR and AHE[26,30]. In[26] the observed anisotropy and change in slope of the Hall response is ascribed to an emergence of magnetization at the interface presumably due to breaking of Kondo singlets. However, the origin of the magnetic impurities leading to a Kondo phase has not been conclusively identified[19,28–31].

The paired non-superconducting regime exists in the same region of carrier density and magnetic field as the reports of AMR and AHE[29]. The pairing field ($B_p$) can vary between 1 T and reach values as high as 15T[21]. $B_p$ is also reported to increase with decreasing carrier density[18], consistent with the dependence of $H_c^{\parallel}$ reported in ref. [26]. Hence, it is natural to ask if the preformed pairing phenomena and 2D AHE share an underlying physical basis.

Another factor influencing the electronic properties of STO-based heterostructures are ferroelastic domains[20,32–35]. Bulk undoped STO undergoes a ferroelastic transition from cubic to tetragonal crystal symmetry at $T \sim 105\mathrm{K}$, leading to the formation of ferroelastic domains[32] which are oriented along the X [100], Y [010] and Z [001] crystalline directions, and separated by nanometer-scale domain walls according to the domain tiling rules[36]. Local probe measurements including scanning SQUID and scanning SET have revealed that transport at the LAO/STO interface is highly inhomogeneous, with current flowing preferentially along the ferroelastic domain boundaries[32,33,35].

Transport measurements on mesoscopic devices created at the LAO/STO interface using conductive atomic force microscope (c-AFM)

lithography[37] provide a powerful platform to explore the rich physics at the interface. Experiments by Pai et al. demonstrated a one-dimensional nature of electron pairing and superconductivity in LAO/STO[38]. The existence of Shubnikov-de Haas like oscillations has been linked to the magnetic depopulation of electron subbands in 1D systems[39] accounting for the widely observed mismatch between Hall carrier density measurements and those revealed by quantum oscillations[25]. The existence of ballistic transport itself in quasi-1D geometries with a mean free path of $\sim 20\mu\mathrm{m}$[19], show signatures which are not obvious from macroscopic 2D measurements but possibly consistent with spatially resolved measurements. If ferroelastic domains, which usually decorate the LAO/STO interface, possess a network of 1D domain walls that percolate in 2D, it is indeed plausible that macroscopic transport behavior might be heavily influenced by the physics of these 1D channels. The high conductance of these edges, which have been demonstrated in numerous experiments, offers a way to connect the mesoscopic physics of quasi-1D devices with the much larger set of experiments performed at macroscopic 2D interface.

Piezoelectric force microscopy imaging of conductive nanowires sketched at the LAO/STO interface using c-AFM lithography reveals that the conducting paths are elongated along the Z-axis at room temperature[40]. Furthermore, low temperature scanning SET measurements of LAO/STO show that while the X and Y ferroelastic domains share similar surface potentials, the Z domains have a higher surface potential, varying by approximately 1meV[32]. Thus, it can be argued that the conductive nanowires created using c-AFM lithography "pre-seed" the formation of Z domains in STO, whereas the X and Y domains define the insulating states.

Here we report mesoscopic transport measurements to probe the correlation between electron pairing, AHE, and electronic nematicity in LAO/STO. These measurements are enabled by quasi-1D cross-shaped ballistic electron waveguides, or "nanocrosses", created at the LAO/STO interface using c-AFM lithography. Ballistic transport in the nanocrosses reveals an electron pairing transition outside the superconducting regime. We find a remarkable agreement between the critical magnetic field above which the electrons de-pair and sharp changes in the Hall response of the nanocrosses, thereby demonstrating a strong correlation between these two distinct phenomena. Angle-dependent Hall measurements further reveal an onset of electronic nematicity that again coincides with the electron pairing transition, unveiling a rotational symmetry breaking due to the transition from paired to unpaired phases at the interface. The emergence of magnetization and nematicity due to the transition of electrons from the paired to unpaired phase shows the significance of preformed pairs on the transport properties in LAO/STO and reveals that these distinct electronic phases are in fact different manifestations of the same underlying physics.

## Results

The LAO/STO samples are grown using pulsed laser deposition details of which are described in the film fabrication section of Methods. The nanocross devices serve as a building block to understand 1D electron physics at the LAO/STO interface. The multi-terminal nature of the nanocross allows four-terminal measurements to be performed simultaneously in both longitudinal and Hall configurations, allowing the two distinct physical phenomena to be directly compared (Fig. 1a). The unique cross shaped geometry of the nanocross defines both Z−X and Z−Y ferroelastic domain boundaries in the system, in close proximity at the nanoscale limit. We have previously discussed the correlation between nanocross devices and ferroelastic domains[20]. Further, the role of angular dependence of electron pairing and AHE is investigated by sculpting nanocross devices at varied angles, $\varphi$ between 0 and 90-degrees with respect to the [100] crystallographic direction (Fig. 1a). All nanocross devices are written at the same location on the sample unless mentioned otherwise.

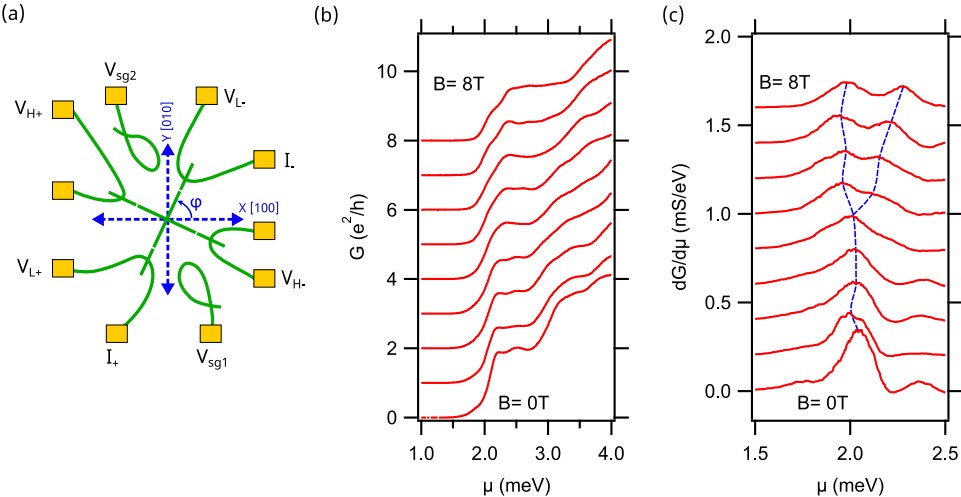

**Fig. 1 | Nanocross device geometry and longitudinal measurements across nanocross Device A1 oriented at $\varphi = 65°$. a** Schematic of longitudinal and Hall transport measurements across the nanocross. Angle $\varphi$ denotes the relative orientation of the nanocross with respect to the crystallographic direction. Longitudinal voltage probes ($V_{L\pm}$) enable four-terminal conductance to be measured while transverse voltage probes ($V_{H\pm}$) enable Hall measurements. Both longitudinal and Hall measurements are acquired simultaneously as a function of gate voltage ($V_{sg1}$ or $V_{sg2}$) and applied magnetic field, $B$, **b** Longitudinal conductance G versus chemical potential $\mu$ for magnetic fields ranging between B = 0T and B = 8T in steps

of 1T for Device A1 oriented at $\varphi = 65°$ with respect to [100] crystallographic direction. A conductance plateau near G≈1.75$e^2$/h appears at all magnetic fields. For magnetic fields larger than B = 4T, the transition to this plateau broadens significantly, and a second plateau is clearly visible at G≈0.90 ± 0.05$e^2$/h at B = 8T. Curves are offset by 1$e^2$/h for clarity, **c** Transconductance dG/dμ versus μ for magnetic fields ranging between B = 0T and B = 8T in steps of 1T for Device A1. dG/dμ versus μ reveals a clear transition between paired and unpaired state near B = 4T as shown by the dashed blue lines. Curves are offset for clarity.

The nanocross geometry, illustrated in Fig. 1a, is composed of two 1 μm-long crossed nanowire segments, with each of the four ends connected to two nanowire leads. Tunnel barriers (see c-AFM lithography section of Methods for details) of width ∼30nm isolate the nanocross from the two terminal leads, allowing the chemical potential to be uniformly tuned by either of the two available proximal side gates at a distance of ∼1μm, from the center of the nanocross, with voltage $V_{sg1}$ and $V_{sg2}$. The precise physical location of the side gates for LAO/STO nanostructures negligibly impacts the electronic structure within the conducting regions[19]. A few volts of back gate voltage $V_{bg}$ is also used to provide coarse tuning of the chemical potential. Four-terminal longitudinal and Hall measurements are performed simultaneously as a function of the applied gate voltage $V_{sg}$ or chemical potential $\mu = \alpha V_{sg}$ where $\alpha$ is the measured lever arm (see Supplementary Fig. 1 for details), and as a function of an applied out-of-plane magnetic field, $\mathbf{B} = B\hat{\mathbf{z}}$. All measurements are performed at or near the base temperature of the dilution refrigerator, $T \sim 50$ mK.

The zero-bias longitudinal conductance $G = dI/dV$ (Fig. 1b) and transconductance $dG/d\mu$ (Fig. 1c and Supplementary Fig. 2c) is shown for Device A1 ($\varphi = 65°$) as a function of $\mu$, for magnetic fields ranging between $B = 0$T and $B = 8$T. Transport is quasi-ballistic with signatures of conductance quantization, similar to reports of straight ballistic electron waveguides[19]. A conductance plateau near $G{\approx}1.75e^2/h$ appears at all magnetic fields. For magnetic fields larger than $B = 4$T, the transition to this plateau broadens significantly, and a second plateau appears at $G{\approx}0.90 \pm 0.05e^2/h$ at $B = 8$T (Fig. 1b). The corresponding line cuts for $dG/d\mu$ (Fig. 1c) focused on the relevant range of chemical potential shows a clear splitting of the $1.75e^2/h$ peak starting at $B = 4$T. The fractional values of the conductance quantization steps is attributed to interference or scattering effects within the nanocross[20].

More insight into the electronic properties of the nanocross can be obtained from examining the transconductance, $dG/d\mu$, as a function of $\mu$ and $B$. Figure 2c shows the transconductance intensity map of Device A1, over the energy range $\mu = 1.90$meV to 2.60meV and the full magnetic field range $-8$T $< B <$ 8T. Analysis of the transconductance peak structure, which is overlaid, reveals a transition from a

single peak to two peaks at a critical field $B_p$. This splitting in transconductance is typical of an electron pairing transition. Fitting of the split peaks above $B_p$ yields an estimate for $B_p \approx 3.9$T $\pm 0.4$T (Fig. 2e).

Next, we focus on the Hall measurements across the nanocross (Fig. 2d). Hall measurements in quasi-1D systems have been widely explored in traditional semiconductors where a quenching of the Hall resistance is observed[41,42]. Hall measurements across quasi-1D nanocrosses is expected to highlight the microscopic origin of 2D Hall measurements reported at the LAO/STO interface. Figure 2f shows the field anti-symmetrized Hall resistance, $R_{xy}^{anti}$ averaged over the energy range $\mu = 1.95$meV to 2.55meV for Device A1. Nonlinearities are observed in the Hall response as a function of magnetic field, $B$. $R_{xy}^{anti}$ vs $B$, shows similar trend at all side-gate potentials $V_{sg}$ (See Supplementary Fig. 3 for details). Fits to the intersection of the low-field and high-field asymptotes (Fig. 2 f) yields a critical value at which the Hall coefficient changes: $B_H \approx 3.4$T $\pm 0.5$T.

Figure 3 shows Hall measurements performed as a function of the orientation of the nanocross with respect to the [100] crystallographic direction (denoted by angle $\varphi$ in Fig. 3a) taken across seven nanocross devices. We investigate two important aspects here: (a) the reproducibility of Hall measurements at a given location and orientation of the nanocross on the sample, and (b) the angular dependence of the Hall response. Figure 3b shows $R_{xy}^{anti}$ averaged over a small range of side-gate voltages for the magnetic field range $-8$T $< B <$ 8T for three nominally identical Devices A1, A2, A3, where the nanocross is oriented at the same angle $\varphi = 65°$ and positioned at the same location on the sample. Nearly identical S-shaped Hall nonlinearities are observed in all three devices (Fig. 3b), in which the Hall coefficient at low magnetic fields is higher than at high magnetic fields. The magnetic field at which the Hall resistance changes slope, labeled as $B_H$, coincides within measurement uncertainty for all three devices A1-A3.

Next, the dependence of the Hall response on the nanocross angle $\varphi$ is summarized in Fig. 3 c. Four distinct angles between 0° and 90° are explored in Devices A1, B1, C, and D, oriented at $\varphi = 65°, 0°, 45°$, and $\varphi = 75°$ respectively. Nonlinear Hall behavior is observed in all devices. However, the shape of the Hall response is found to depend strongly on nanocross orientation.

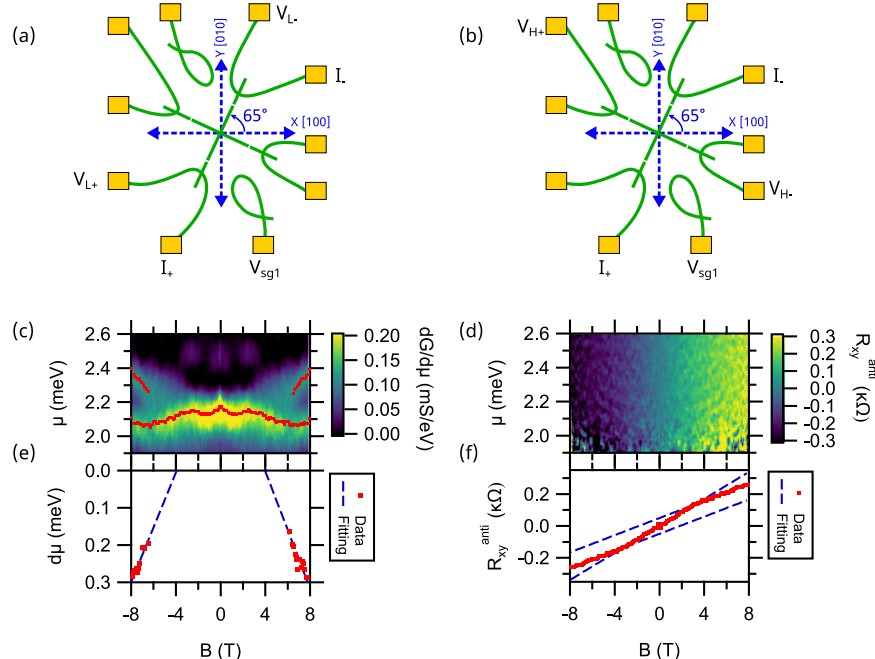

**Fig. 2 | Comparison of transconductance $dG/d\mu$ and Hall measurements across Device A1 oriented at $\varphi = 65°$. a, b** Schematic showing the current and voltage lead configurations for longitudinal and Hall measurement across the nanocross. **c** Intensity plot of transconductance $dG/d\mu$ versus chemical potential $\mu$ and magnetic field $B$. Fits to peak of the transconductance versus magnetic field are overlaid. The splitting in the transconductance from a single peak to two peaks is characteristic of the electron pairing transition. **d** Intensity plot of anti-symmetrized Hall resistance $R_{xy}^{anti}$ versus $\mu$ and $B$. **e** Plot of energy difference between transconductance peaks versus magnetic field. Blue dashed line extrapolates to a value of $B_P = 3.9 \pm 0.4T$. **f** Average $R_{xy}^{anti}$ over the range $\mu = 1.95meV$ to $2.55meV$ reveals nonlinear behavior with asymptotes that cross at $B_H = 3.4 \pm 0.5T$.

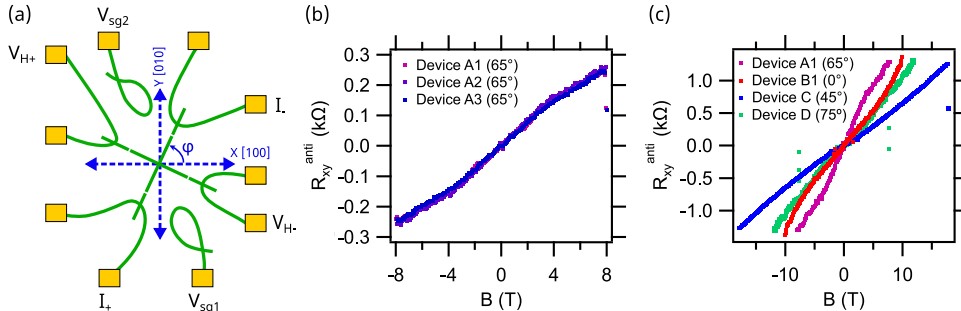

**Fig. 3 | Angle dependence of anomalous Hall response. a** Schematic showing the current and voltage leads for Hall measurement across the nanocross. Angle $\varphi$ denotes the relative orientation of the nanocross devices with respect to the [100] crystallographic direction. **b** Hall measurements across Device A1, A2, A3 oriented at $\varphi = 65°$ with respect to the [100] crystallographic direction. The Hall resistance overlaps within the uncertainty of measurement. **c** Variation of Hall resistance as a function of $\varphi$. Hall measurements across nanocross devices A1, B1, C and D oriented at $\varphi = 65°$, $0°$, $45°$, and $75°$ respectively. Hall resistance for Device A1, B1 and D is amplified by 5x, 2.5x and 1.5x respectively for clarity.

Table 1 summarizes the low-field Hall coefficient, $R_H^{low}$ and high-field Hall coefficient, $R_H^{high}$, with the separation between low and high being $B_H$, obtained from experiments with eight devices. Fitting procedures and additional Hall measurements for devices with $\varphi = 0°$ and $\varphi = 65°$ are shown in Supplementary Figs. 4 and 5. Interestingly, all devices exhibit comparable values of Hall slope when $|B| < B_H$. The Hall transition field, $B_H$ is minimum for $\varphi = 45°$, $B_H \sim 1.8T \pm 0.2T$ and maximum for $\varphi = 0°$, $B_H \sim 5.9T \pm 0.1T$ within the range of error. The Hall slope for Device B2 could not be identified with the same degree of accuracy in the high field regime but $B_H \sim 5.2T$ for magnetic field range, $-7T < B < 7T$ (Supplementary Fig. 4f) similar to Device B1 also written at $\varphi \sim 0°$.

The pairing transition in electron waveguides is characterized by a new conductance plateau at $G = e^2/h$ between $G = 0$ and $G = 2e^2/h$, which takes place for $|B| = B_p$. For $|B| < B_p$, transport is governed by a single quantum channel composed of electron pairs that propagate quasi-ballistically. For $|B| > B_p$, the paired channel splits into spin-up and spin-down single-electron channels with subband bottoms that split in energy and appear as two distinct peaks on the transconductance (Fig. 2c). The experimental results show that the field at which the Hall slope changes ($B_H = 3.4 \pm 0.5T$) coincides, within error, with electron pairing transition ($B_p = 3.9 \pm 0.4T$). This relationship holds regardless of the angle of the nanocross with respect to the crystallographic direction and value of the pairing field. Results on Device E Sample 2 for $\varphi = 45°$, summarized in Supplementary Figs. 6 an. 7 yield $B_p = 2.2 \pm 0.4T$ (Supplementary Fig. 7e) and $B_H = 2.4 \pm 0.6T$ (Supplementary Fig. 7f), which again agree within the uncertainty of measurement.

The striking agreement between the pairing field and anomalous Hall response in LAO/STO heterostructures across nanocross devices

sculpted on two different samples at different orientations with respect to the [100] crystallographic direction and pairing field differing by a factor of two suggest an underlying physical mechanism that relates them. Previous explanations for anomalous Hall response were mainly restricted to single-particle descriptions involving multiple bands[17,31,43] or invoked magnetic interactions of unknown origins[26,30]. In ref. 26 the anomalous Hall signature is described as a metamagnetic transition, an "emergence of magnetization" which occurs at a critical magnetic field in 2–15 Tesla range. Our experimental findings point to a specific origin of excess magnetization, one which is associated with the breaking of spin-singlet electron pairs. Above the pairing field, spin-singlet electron pairs unbind and spin-polarize, resulting in characteristic changes in the Hall response. This scenario was postulated in[29] but lacked a strong empirical basis. The results reported here provide direct evidence in support of this mechanism, associating the AHE with the pairing transition.

To understand the angular dependence of the Hall resistance across the nanocross[20], we need to consider the role of ferroelastic

**Table 1 | Hall transition field, $B_H$, electron pairing field, $B_P$, and slope of anomalous Hall response below ($R_H^{low}$) and above ($R_H^{high}$) the transition field $B_H$, summarized for eight nanocross devices A1-E oriented for $0° < \varphi < 90°$**

| Nanocross angle φ | Sample | Device | $B_P$ (T) | $B_H$ (T) | $R_H^{low}$ (Ω/T) | $R_H^{high}$ (Ω/T) |
|---|---|---|---|---|---|---|
| 65° | 1 | A1 | 3.9 ± 0.4 | 3.4 ± 0.5 | 43 | 26 |
| | | A2 | | 3.3 ± 0.4 | 42 | 24 |
| | | A3 | | 3.2 ± 0.3 | 42 | 24 |
| 0° | | B1 | | 5.9 ± 0.1 | 43 | 74 |
| | | B2 | | 5.2 ± 2.9 | 44 | 37 |
| 45° | | C | | 1.8 ± 0.2 | 49 | 70 |
| 75° | | D | | 2.2 ± 0.1 | 49 | 76 |
| 45° | 2 | E | 2.2 ± 0.4 | 2.4 ± 0.6 | 44 | 32 |

domains and their connection to the relevant d-orbital bands at the LAO/STO interface. Prior magnetotransport measurements on nanocross devices have revealed an inhomogeneous energy landscape which is nevertheless highly reproducible from one device to another[20] (similar to the three devices A1-A3 from Fig. 3). The observed inhomogeneity[20] is attributed to a highly reproducible ferroelastic domain configuration artificially described at the LAO/STO interface by the nanocross devices. We extend this ferroelastic domain model provided in[20] for the four angles of the nanocross discussed earlier (Fig. 4e–h). For simplicity we only consider the lowest energy configuration. A clear variation is observed in the ferroelastic domain configuration, with $\varphi = 0°$ (Fig. 4e) and $\varphi = 45°$ (Fig. 4h) configurations forming the two extremes. While $\varphi = 0°$ signifies that the nanocross naturally coincides with the crystallographic axis, $\varphi = 45°$ has the nanocross aligned parallel to the X-Y domain boundary, and the nanocrosses with $\varphi = 65°$ and $\varphi = 75°$ are intermediate between the two extreme configurations. The observed minima and maxima of the Hall transition field, $B_H$ also coincides with the two extreme ferroelastic domain configurations as summarized in Fig. 4 and Table 1. As mentioned earlier, the pairing field in mesoscopic devices has been found to vary between 2 T and 15 T[21–23]. The possible role of ferroelastic domains and domain boundaries in mediating electron pairing in LAO/STO-based nanostructures has also been previously suggested in ref. 38. The results presented here across nanocross devices at varied angles with respect to the crystallographic direction give further empirical evidence linking the preformed electron pairs, AHE and ferroelastic domain structures in LAO/STO.

Angle dependence of Hall resistance shares an important aspect with AHE and AMR studies reported in literature[25]. Above a critical magnetic field, they all exhibit a dramatic change in anisotropy or nonlinearity in the transport properties at the interface. Figure 5a shows the variation of $R_{xy}^{anti}$ as the magnetic field strength is increased from 1T to 7T for $0° \leq \varphi \leq 180°$. The graph assumes two axes of symmetry, rotational symmetry by 90° and mirror symmetry along 45°, and is interpolated between measured values. Figure 5a reveals the

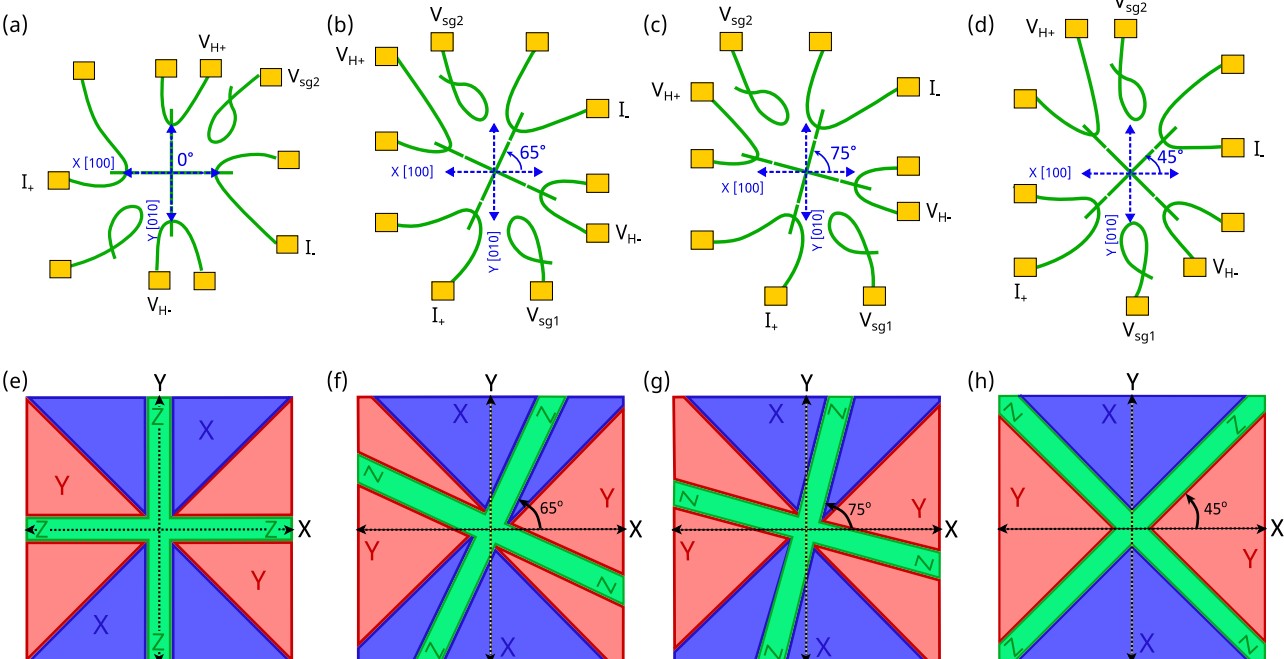

**Fig. 4 | Ferroelastic domain model for nanocross devices oriented at 0°<φ<90° with respect to the [100] crystallographic direction. a−d** Schematic showing the current and voltage leads for Hall measurement across the nanocross devices oriented at $\varphi = 0°$, 65°, 75° and 45° respectively. **e−h** The domain configuration of a symmetric nanocross in the lowest energy configuration for devices oriented at $\varphi = 0°$, 65°, 75° and 45° respectively. The Z-X, Z-Y and X-Y domain boundaries have been defined by darker shades along the edges of the nanocross for all cases.

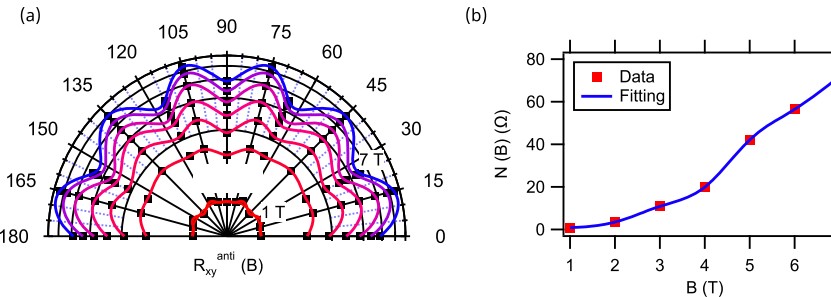

**Fig. 5 | Angle dependence of Hall response and electron nematicity in LAO/STO at increasing magnetic field strengths.** Angle $\varphi$ denotes the relative orientation of the nanocross with respect to the [100] crystallographic direction. **a** Spline fit overlaid with the experimental data points show the variation in $R_{xy}^{anti}$ with increasing magnetic field strength, $1T \leq B \leq 7T$ for $0 \leq \varphi \leq 180°$. The graph considers two axes of symmetry, rotational symmetry by 90° and mirror symmetry along 45°. **b** Evolution of nematicity marker $N(B) = \left\langle \Delta R_{xy}^{anti2} \right\rangle^{1/2}$ as a function of magnitude of magnetic field $B$.

increase in variation of $R_{xy}^{anti}$ vs $\varphi$ with increasing magnitude of magnetic field. A non-monotonic trend is observed in $R_{xy}^{anti}$ versus $\varphi$ with higher harmonic in $\varphi$ as previously reported for in-plane AMR measurements[26–28,44,45]. Additionally, Supplementary Fig. 8 shows the increase in variation of the Hall coefficient, $R_H$ vs $\varphi$ with increasing magnitude of magnetic field and in-plane anisotropy of critical field $B_H$ vs $\varphi$.

To quantify the non-monotonic behavior of $R_{xy}^{anti}$, we define a measure of nematicity, $N(B)$ as the standard deviation of $R_{xy}^{anti}$ over the interval $0° < \varphi < 90°$. For an isotropic system, $N(B)$ is expected to be close to zero. However, Fig. 5b shows that as the magnitude of magnetic field increases, there is an increase in the angular variation of the Hall response, which is quantified by the increasing magnitude of $N(B)$ indicating the onset of electronic nematicity.

As mentioned previously, AMR and emergence of nonlinearities in Hall resistance at a critical magnetic field $B_H$ has often been linked to a Lifshitz transition in which electrons from the $d_{xz}$ and $d_{yz}$ bands contribute to transport, in addition to the lower-energy $d_{xy}$ band[17,31]. The preformed pairs which are most likely composed of isotropic $d_{xy}$ carriers, dominate the low-field normal Hall response. The onset of anisotropic transport above the pairing field suggests that when electrons de-pair, they acquire $d_{xz}/d_{yz}$ characteristics which are known to be highly anisotropic. The pairing transition is thus consistent with the Lifshitz picture, but with a shift in electron orbitals coinciding with the pairing transition itself. This scenario also provides a plausible explanation for the consistent value of $R_H^{low}$ for all eight devices (see Table 1), since they are derived from the Hall response of the preformed $d_{xy}$ electron pairs. The value of $R_H^{low}$ presented here (see Table 1), also closely matches with the value of $R_H^{low}$ ($\approx 40\Omega/T$) reported in[26] for 2D Hall bars with magnetic field applied out of plane.

In summary, simultaneous longitudinal and Hall measurements on novel quasi-1D ballistic nanocrosses sketched at the LAO/STO interface show a direct correlation between the electron pairing transition and nonlinearities in the Hall response. The correlation between electron pairing and AHE is independent of the orientation of the nanocross with respect to the crystallographic direction and of the magnitude of electron pairing field. Angle-dependent Hall measurements taken across multiple nanocross orientations further show evidence of electronic nematicity whose onset also coincides with the pairing transition. A natural explanation is connecting the electron pairing transition to a shift between $d_{xy}$ electron pairs and $d_{xz}$ and $d_{yz}$ unpaired states, with the latter exhibiting a high degree of anisotropic behavior. The correlation between electron pairing, AHE and electronic nematicity consolidates a wide range of seemingly disparate experimental findings reported in STO and construct a comprehensive understanding of the rich correlated nanoelectronics present in this system. The results presented in this work provide several new insights regarding this system; the role of ferroelastic domains as elusive

pairing "glue" in STO and the importance of the 1D paired liquid phase, in general the pre-formed pairs on the transport properties in LAO/STO. Although the existence of the paired liquid states at the LAO/STO interface has been known for half a decade now, they still fail to find a place in the phase space of STO-based heterostructures. The given results reinforce the need to go beyond the single-particle descriptions and consider these pre-formed pairs as an essential element of the phase diagram of STO-based systems. These results can possibly be extended to other correlated systems and non-conventional superconductors where pre-formed pairs are known to exist but not considered while studying the transport phenomena.

## Methods
### Film fabrication
The 3.4-unit cell LAO/STO samples are epitaxially grown on TiO$_2$-terminated STO (001) substrates using pulsed laser deposition. The thickness of LAO is precisely controlled by in-situ RHEED monitoring. To make a TiO$_2$-terminated substrate, as received STO substrates are etched with buffered HF for 1 min and annealed at 1000°C for six hours. During the LAO growth, the substrate temperature is kept at 550°C and oxygen partial pressure is $10^{-3}$ mbar. LAO target is focused by KrF (248nm) excimer laser at a repetition rate of 3Hz and a fluence of $1.8 J/cm^2$. After growth, the sample is slowly cooled down to room temperature under oxygen pressure of 1atm.

### c-AFM lithography
Sixteen interface contacts, formed by milling 25 nm-deep trenches and subsequently depositing Ti/Au (4 nm/25 nm), surround a 25 μm × 25 μm "canvas" where devices are "sketched" with a voltage-biased c-AFM tip at ambient temperature. Conducting paths are created by applying a positive bias $V_{tip} \sim 10$V to the AFM tip, which locally protonates the LAO surface, thereby rendering the interface locally $n$-type conductive. An insulating state is locally restored by applying negative voltages to the tip ($V_{tip} \sim -3$V). The nanocross is composed of two 1 μm-long crossed nanowire segments, created using a positive tip voltage $V_{tip} = 12$V. Each arm of the nanocross has a tunnel barrier which is created by "erasing" with a negative tip voltage $V_{tip} = -4$V over a distance $w_b = 30$nm. The tunnel barriers decouple the nanocross from the two terminal leads, allowing the electron density of the nanocross to be tuned by a proximal side-gate, $V_{sg}$. Conductive nanostructures created by c-AFM lithography have a 2D carrier density typically in the range of $0.5-1.0 \times 10^{13}$ $cm^{-2}$ and a 2D electron mobility $\mu_H \sim 10^3$ $cm^2/(V\,s)$.

## Data availability
All data generated and analyzed during this study are included in this article and its supplementary information and are available from the corresponding author upon request.

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

## Acknowledgements

We thank Bharat Jalan and Beena Kalisky for helpful discussions. CBE acknowledges support for this research through the Gordon and Betty Moore Foundation's EPiQS Initiative, Grant GBMF9065 and

a Vannevar Bush Faculty Fellowship (ONR N00014-20-1-2844). Transport measurement at the University of Wisconsin–Madison was supported by the US Department of Energy (DOE), Office of Science, Office of Basic Energy Sciences (BES), under award number DE-FG02-06ER46327. J.L. acknowledges support from NSF DMR-2225888 and AFOSR FA9550-23-1-0368.

## Author contributions

A.N. and J.L. conceived the idea, designed the experiment, and prepared the manuscript. A.N. led and J.L. supervised the project. H.L., K.E., and C.B.E. prepared the samples. J.Li. processed and patterned the samples. A.N. performed experiments and analyzed data with support from M.B. and Y.Y.P. A.N. and P.I. designed data acquisition and analysis software. All authors discussed the results and revised the manuscript.

## Competing interests

The authors declare no competing interests.
