## [Peer Review File · Nature Communications]

REVIEWER COMMENTS

Reviewer #1 (Remarks to the Author):

The manuscript entitled “Electron Pairing and Nematicity in LaAlO₃/SrTiO₃ Nanostructures” addresses the origin of non-linearity in the magnetic field dependence of the Hall resistivity above a critical field in two-dimensional electron gas stabilized at the interface of SrTiO₃ with certain other perovskite oxides. This is a ubiquitous observation that has attracted several interpretations ranging from a simple and realistic two-band conduction originating in the d_{xy} and d_{yz}/d_{zx} derived states, to more exotic interpretations such as formation of Kondo singlets and the stabilization of a magnetic order beyond a critical field, which has been identified as B_H in this manuscript. This manuscript also attributes the non-linearity to the appearance of a magnetic order, indirectly through the measurements of longitudinal conductance of just two samples at a fixed temperature (50 mK) and a fixed value of backgate voltage, which critically decides the carrier density (n) in the channel and hence the operating point in the n - T phase space. The key result of this study is the observation of an additional plateau in the conductance (G) of the sample at $G \sim e^2/h$ above a certain field B_P which coexists with the conductance plateau at $\sim 2e^2/h$. The authors attribute this new feature to breaking of preformed singlet pairs in the gas which are awaiting condensation. One would like to know if the B_P is somehow connected to the Pauli limit for this system. It is argued that the carriers from the broken singlet pairs spin polarize, and thus, lead to the onset of a magnetic order. For the two samples investigated, the B_P matched the critical field B_H deduced from Hall measurements. The authors also argue that at field $B > B_H$ (B_P), the transport is anisotropic on the (001) plane of the interface. While in words this picture appears quite attractive, it is not supported by measurements on a larger set of samples, larger values of the angle θ and at different points of the n - T phase space, and therefore, is not suitable for a priority publication. The following additional observations support this conclusion.

1. There are only two data points available for B_P taken from measurements on two different samples at a fixed value of the orientation of the conducting channel with respect to the principal axes of the crystal ((100) or (010)). Further, the tinny feature in the G vs θ plots of Fig. 1(b) around which the entire message of this manuscript is built, is barely visible in data taken at 7 and 8 tesla. There are many other prominent features in the data of Fig. 1(b) at the higher value of θ . While the derivative plot of Fig. 1(c) is drawn over a very limited range of θ , the readers would wonder about the origin of such high energy features, which have been conveniently ignored.
2. Although the Hall data have a larger spread, one would like to see the angular dependence of the Hall slope and the of critical field at which deviation from linearity occur as a function of angle θ as it is changed from zero to 90 degrees by sculpting several nanochannels. One would also expect the zero and 90-degree result to be degenerate.
3. The experimental section does not provide any information about device fabrication. For example, the temperature at which the nanochannels were written is not mentioned. If these structures were sculpted at ambient temperature, then one cannot say what will be their position with respect to

the ferroelastic domains as their appearance is a stochastic process occurring at the phase change point. The literature on ferroelastic domains also suggests that the domain size could be much larger than the size of the device. The domain patterns at the surface of the crystal may also have several orientations (see, for example, J Appl. Phys. 86, 1653 (1999)).

4. Supplementary information on the quality of the samples is missing. One would like to know the number of LAO monolayers, concentration, and mobility of charge carriers in the 2D gas, and the propensity of this gas to undergo condensation. Would the sample become superconducting if appropriate gate bias is used?

5. Lastly, the classic paper on superconductivity in reduced SrTiO₃ crystals (Ref. # 14) could have been referenced in a better context. It does not say anything about two-dimensional superconductivity or heterointerfaces. This study is on bulk reduced crystals, and in many respects seems to tell us why LAO/STO interface becomes superconducting on high temperature annealing and on bombardment by energetic species ejected from the target during growth.

Reviewer #2 (Remarks to the Author):

This manuscript by the group of Jeremy Levy reports on a study of magneto transport in crossbar nanodevices of LaAlO₃/SrTiO₃ interface 2DEGs. The main finding of this work is a correlation between the onset of magnetic field-dependent anomalies in the Hall effect and the onset of strong angle dependence in the magnetoresistance which shows structure indicative of a Fermi surface created from underlying highly anisotropic orbitals.

Their results suggest a pair-breaking induced density reorganization between d_{xy} and d_{xz}/d_{yz} orbitals as a function of the magnetic field.

The term 'nematicity' is not correctly applied here since in a crystal this terminology must refer to a state where some underlying crystal symmetry is spontaneously broken. Principally, the presence of angular variations in the field dependent AMR does not imply nematic order which is what the authors seem to be claiming since it is not clear that the crystalline C₄ symmetry is being broken to C₂.

The second issue is that much of the information upon which this claim is based is not new.

(i) There has been an old observation of anomaly in the AMR by the Ilani group back in 2013 (ref.24) at $B \sim 3T$ where crystalline components become visible. This was identified as being possibly due to

new orbitals becoming important - speculated but not substantiated as being via a lattice Kondo effect.

(ii) Several previous works by J. Levy group has already shown that there are conductance plateaus in LAO/STO nanostructures whose details (conductance values, plateau widths, etc) depend on the participating sub-bands and device geometry.

(iii) The main observation here seems to be Fig.1c where a splitting of a central peak at $\mu \sim 2\text{meV}$ is seen for $B \sim 3\text{T}$. However, taken by itself, it is not clear that this implies that it is a pairing gap (presumably from dxy orbital based on the manuscript) evolving into Zeeman split gap as opposed to, say, a Kondo hybridization gap. In addition, orbital dependent peaks are not seen in this figure. If additional orbitals are becoming populated or depopulated at this field, why are no additional features seen associated with them around these energies? Furthermore, the data is shown for a single device and not for various devices and at various temperatures to show that this field scale is correlated in AMR and in these tunneling cross-conductance.

Overall, I think the paper makes an interesting but still very speculative suggestion. If there was extensive new data or new physics being discovered it might have been more reasonable to consider this for Nature Communications.

Reviewer #3 (Remarks to the Author):

The study conducted by Aditi Nethewala and colleagues explores the transport properties of quasi-1D structures on SrTiO₃/LaAlO₃ interfaces fabricated using a conducting atomic force microscope. The authors establish correlations between conductance versus gate voltage, Hall signal, and direction dependence of the Hall effect to uncover the underlying transport mechanisms.

The authors interpret the correlation between the field at which the Hall becomes anisotropic and the vanishing of features in the conductance characteristics as evidence for the importance of

performed pairs in the transport properties of STO-Based heterostructures. The authors claim that a single energy scale governs the transport features of these interfaces.

While some of the features have been reported, for example, Figure 1 shows data similar to previous publications from the same group (e.g., Ref 17), the new merit of this contribution is the correlation between the features.

While the findings are noteworthy, some concerns remain. These issues require clarification before publication in Nature Communications.

A potential issue with the argument put forth by Nethewala et al. is the apparent temperature dependence of various transport features, as seen in previous publications and cited within this study. Specifically, the anomalous Hall and anisotropic magnetoresistance have been observed to persist up to approximately 30 K, which is a characteristic temperature associated with the formation of ferroelastic polar domains in SrTiO₃. It is, therefore, unclear whether the observed pairs that condense at 300 mK could form at temperatures two orders of magnitude higher. It would be valuable for the authors to demonstrate whether the depairing field, nematicity, and anomalous Hall exhibit the same temperature dependence to clarify this matter.

I do not understand why the Hall slope in figure S2 does not change with gate voltage. Doesn't that mean that the side gates are ineffective in changing the carrier density?

Due to the nonlinear dielectric constant, the chemical potential, μ , is not necessarily proportional to the gate voltage. The lever arm model is, I believe, relevant for the linear response, which does not apply to STO

In figure 5, it is not clear what the fit is based on, There are only a few data points, if I understand correctly, yet the fit is very detailed. Can the authors show the data the fit is based on?

After the authors have thoroughly addressed the aforementioned concerns, I will be in a position to recommend the publication of their research in Nature Communications.

RESPONSE DOCUMENT

We thank all the Reviewers for their constructive comments. The Reviewer comments are in this style and *our response is in this style. Changes implemented in the revised version of the manuscript are in this style.*

Reviewer #1 (Remarks to the Author):

The manuscript entitled “Electron Pairing and Nematicity in LaAlO₃/SrTiO₃ Nanostructures” addresses the origin of non-linearity in the magnetic field dependence of the Hall resistivity above a critical field in two-dimensional electron gas stabilized at the interface of SrTiO₃ with certain other perovskite oxides. This is a ubiquitous observation that has attracted several interpretations ranging from a simple and realistic two-band conduction originating in the d_{xy} and d_{yz}/d_{zx} derived states, to more exotic interpretations such as formation of Kondo singlets and the stabilization of a magnetic order beyond a critical field, which has been identified as BH in this manuscript.

This manuscript also attributes the non-linearity to the appearance of a magnetic order, indirectly through the measurements of longitudinal conductance of just two samples at a fixed temperature (50 mK) and a fixed value of backgate voltage, which critically decides the carrier density (n) in the channel and hence the operating point in the n-T phase space.

We thank the referee for the detailed feedback and comments, which we address below.

The key result of this study is the observation of an additional plateau in the conductance (G) of the sample at $G \sim e^2/h$ above a certain field B_P which coexists with the conductance plateau at $\sim 2e^2/h$. The authors attribute this new feature to breaking of preformed singlet pairs in the gas which are awaiting condensation. One would like to know if the B_P is somehow connected to the Pauli limit for this system.

We thank the referee for asking this question, which was not explicitly addressed in the manuscript.

Pauli limit violation has been previously reported in LAO/STO (24). However, in the given manuscript, we are discussing about electron pairing without superconductivity as opposed to properties of the superconducting state and hence the Pauli limit does not apply.

We have pointed this fact out in the revised manuscript.

Lines 53-56: “The characteristic magnetic field, B_p at which electrons unbind can be two orders of magnitude larger than the boundary for superconductivity ($B_p \gg B_c \sim 0.2 \text{ T}$) and the paired electron states are stable at temperatures as high as 900 millikelvin (mK), well above the superconducting transition temperature ($T_c \sim 300 \text{ mK}$) where the Pauli limit does not apply (24).”

It is argued that the carriers from the broken singlet pairs spin polarize, and thus, lead to the onset of a magnetic order. For the two samples investigated, the BP matched the critical field B_H deduced from Hall measurements. The authors also argue that at field $B > B_H$ (BP), the transport is anisotropic on the (001) plane of the interface. While in words this picture appears quite attractive, it is not supported by measurements on a larger set of samples, larger values of the angle ϕ and at different points of the n-T phase space, and therefore, is not suitable for a priority publication.

We appreciate the Referee expressing concern about the number of devices, which can always be expanded with more investigation. Our results which clearly show the anisotropic nature of transport or electron nematicity at the LAO/STO interface is supported by data taken over seven devices (Devices A1-A3, B1-B2, C, D, Sample 1) for multiple values of angle $\phi = 0^\circ, 45^\circ, 65^\circ$, and 75° with respect to the [100] crystallographic direction. The characteristic magnetic field, B_H at which the Hall coefficient changes is also shown to be highly reproducible for nominally identical nanocross devices written at the same location and orientation on the sample during different conductive atomic force microscope lithography cycles. Further, the direct correlation between electron pairing and anomalous Hall effect is shown for two different orientations of the nanocross device with respect to the [100] crystallographic direction ($\phi = 65^\circ$ for Device A1 and $\phi = 45^\circ$ for Device E) and pairing field B_p differing by a factor of 2 ($B_p = 3.4 \pm 0.4 T$ and $2.2 \pm 0.4 T$ respectively). The correlation between electron pairing, anomalous Hall effect and electron nematicity shown over a large set of devices with significantly different pairing field and multiple values of angle ϕ is strong evidence to support our central claim.

Concerning the question about the n-T phase space, such an investigation would not be possible since quantized ballistic transport, one of the key quantities being measured, is not measurable at significantly elevated temperatures. The electronic subbands will get blurred at higher temperatures and hence one cannot resolve the pairing field, B_p .

Further, unlike 2D Hall bars where large backgate voltages are applied to tune the carrier density of the system, the nanocross devices are composed of two $1 \mu\text{m}$ -long crossed nanowire segments isolated from the two terminal leads by four highly transparent tunneling barriers. These tunneling barriers are very sensitive to the applied backgate voltage. Only a small window of backgate voltage (within a few millivolts) is available to coarsely tune all the four tunneling barriers.

We have modified the main text to highlight the important details regarding the larger set of samples studied over multiple values of angle ϕ and the reproducibility of our results to support our conclusions at several instances throughout the revised version of the manuscript.

Lines 126-128: “Further, the role of angular dependence of electron pairing and AHE is investigated by sculpting nanocross devices at varied angles, φ between 0 and 90-degrees with respect to the [100] crystallographic direction (Figure 1(a)).”

Lines 173-177: “Figure 3 shows Hall measurements performed as a function of the orientation of the nanocross with respect to the [100] crystallographic direction (denoted by angle φ in Figure 3 (a)) taken across seven nanocross devices. We investigate two important aspects here: (a) the reproducibility of Hall measurements at a given location and orientation of the nanocross on the sample, and (b) the angular dependence of the Hall response.”

Lines 210-213: “The striking agreement between the pairing field and anomalous Hall response in LAO/STO heterostructures across nanocross devices sculpted on two different samples at different orientations with respect to the [100] crystallographic direction and pairing field differing by a factor of two suggest an underlying physical mechanism that relates them.”

Lines 239-242: “The results presented here across nanocross devices at varied angles with respect to the crystallographic direction give further empirical evidence linking the preformed electron pairs, AHE and ferroelastic domain structures in LAO/STO.”

Lines 275-279: “The correlation between electron pairing and AHE is independent of the orientation of the nanocross with respect to the crystallographic direction and of the magnitude of electron pairing field. Angle-dependent Hall measurements taken across multiple nanocross orientations further show evidence of electronic nematicity whose onset also coincides with the pairing transition.”

The following additional observations support this conclusion.

1. There are only two data points available for BP taken from measurements on two different samples at a fixed value of the orientation of the conducting channel with respect to the principal axes of the crystal ((100) or (010)).

We repeated these experiments to verify that the measurements are reproducible, which they are. We also investigated the orientation dependence. The direct correlation between electron pairing and anomalous Hall effect at the LAO/STO interface is shown for two different orientations of the conducting channel ($\varphi = 65^\circ$ for Device A1 Sample 1 and $\varphi = 45^\circ$ for Device E Sample 2) with respect to the [100] crystallographic direction (Figure 2 and Figure S7 of the manuscript, respectively; also shown below). Further, the pairing fields B_p differs by a factor of 2 for the representative cases. We believe that the observation of a direct correlation between electron pairing and anomalous Hall effect on two different samples at two nanocross orientations and for significantly different electron pairing fields signifies that the given correlation is a robust phenomenon.

We have modified the main text and figure captions (as shown below) to highlight the different orientations of the conducting channel in the revised manuscript.

Figure 2: Comparison of transconductance $dG/d\mu$ and Hall measurements across Device A1 oriented at $\varphi = 65^\circ$. (a, b) Schematic showing the current and voltage lead configurations for longitudinal and Hall measurement across the nanocross. (c) Intensity plot of transconductance $dG/d\mu$ versus chemical potential μ and magnetic field B . Fits to peak of the transconductance versus magnetic field are overlaid. The splitting in the transconductance from a single peak to two peaks is characteristic of the electron pairing transition. (d) Intensity plot of anti-symmetrized Hall resistance R_{xy}^{anti} versus μ and B . (e) Plot of energy difference between transconductance peaks versus magnetic field. Blue dashed line extrapolates to a value of $B_p = 3.9 \pm 0.4 T$. (f) Average R_{xy}^{anti} over the range $\mu = 1.95 meV$ to $2.55 meV$ reveals nonlinear behavior with asymptotes that cross at $B_H = 3.4 \pm 0.5 T$.

Figure S7: Comparison of transconductance $dG/d\mu$ and Hall measurements on Device E sample 2 oriented at $\varphi = 45^\circ$. (a, b) Schematic showing the current and voltage lead configurations for longitudinal and Hall measurement across nanocross device E. (c) Intensity plot of transconductance $dG/d\mu$ versus chemical potential μ and magnetic field B . Fits to peak of transconductance versus magnetic field are overlaid. The splitting in the transconductance from a single peak to two peaks is characteristic of the electron pairing transition. (d) Intensity plot of Hall resistance R_{xy}^{anti} versus μ and B . (e) Plot of energy difference between transconductance peaks versus magnetic field. Blue dashed line extrapolates to a value of $B_p = 2.2 \pm 0.4$ T. (f) Average Hall resistance over the range $\mu = 0.51$ meV to 0.52 meV reveals nonlinear behavior with asymptotes that cross at $B_H = 2.4 \pm 0.6$ T.

Lines 126-128: “Further, the role of angular dependence of electron pairing and AHE is investigated by sculpting nanocross devices at varied angles, φ between 0 and 90-degrees with respect to the [100] crystallographic direction (Figure 1 (a)).”

Lines 210-213: “The striking agreement between the pairing field and anomalous Hall response in LAO/STO heterostructures across nanocross devices sculpted on two different samples at different orientations with respect to the [100] crystallographic direction and pairing field differing by a factor of two suggest an underlying physical mechanism that relates them.”

Lines 275-279: “The correlation between electron pairing and AHE is independent of the orientation of the nanocross with respect to the crystallographic direction and of the magnitude of electron pairing field. Angle-dependent Hall measurements taken across multiple nanocross

orientations further show evidence of electronic nematicity whose onset also coincides with the pairing transition.”

Further, the tinny feature in the G vs m plots of Fig. 1(b) around which the entire message of this manuscript is built, is barely visible in data taken at 7 and 8 Tesla.

We understand the concern of the Referee. However, the identification of these subtle features is based on well-established physics. The transition of the electrons from the paired state to the unpaired state is highlighted by the blue dashed lines in the transconductance versus chemical potential plot shown in Figure 1c. Further, analysis of the transconductance peak structure overlaid with the transconductance intensity map (Figure 2c) also reveals the transition from a single peak to two peaks at the pairing field B_p . This splitting in the transconductance is characteristic of the electron pairing transition. For $|B| < B_p$, transport is governed by a single quantum channel consisting of electron pairs that propagate quasi-ballistically. For $|B| > B_p$, the paired channel splits into spin-up and spin-down single-electron channels, with subband bottoms that split in energy and show up as two distinct peaks in the transconductance map. The analysis shown in Figure 1c and Figure 2c further supports the observation of the $G \approx 0.90 \pm 0.05 e^2/h$ plateau for $|B| > 4T$ as first highlighted in Figure 1b. We have included the important results showing the clear transition of electrons from paired to unpaired state at $|B| > 4T$ below.

Figure 1: Nanocross device geometry and longitudinal measurements across nanocross Device A1 oriented at $\varphi = 65^\circ$. (a) Schematic of longitudinal and Hall transport measurements across the nanocross. Angle φ denotes the relative orientation of the nanocross with respect to the crystallographic direction. Longitudinal voltage probes ($V_{L\pm}$) enable four-terminal conductance to be measured while transverse voltage probes ($V_{H\pm}$) enable Hall measurements. Both longitudinal and Hall measurements are acquired simultaneously as a function of gate voltage (V_{sg1} or V_{sg2}) and applied magnetic field, B , (b) Longitudinal conductance G versus chemical potential μ for magnetic fields ranging between $B = 0$ T and $B = 8$ T in steps of 1 T for Device A1 oriented at $\varphi = 65^\circ$ with respect to [100] crystallographic direction. A conductance plateau near $G \approx 1.75 e^2/h$ appears at all magnetic fields. For magnetic fields larger than $B = 4$ T, the transition to this plateau broadens significantly, and a second plateau is clearly visible at $G \approx 0.90 \pm 0.05 e^2/h$ at $B = 8$ T. Curves are offset by $1 e^2/h$ for clarity, (c) Transconductance $dG/d\mu$ versus μ for magnetic fields ranging between $B = 0$ T and $B = 8$ T in steps of 1 T for Device A1. **$dG/d\mu$ versus μ reveals a clear transition between paired and unpaired state near $B = 4$ T as shown by the dashed blue lines.** Curves are offset for clarity.

Figure 2: Comparison of transconductance $dG/d\mu$ and Hall measurements across Device A1 oriented at $\varphi = 65^\circ$. (a, b) Schematic showing the current and voltage lead configurations for longitudinal and Hall measurement across the nanocross. (c) Intensity plot of transconductance $dG/d\mu$ versus chemical potential μ and magnetic field B . Fits to peak of the transconductance versus magnetic field are overlaid. **The splitting in the transconductance from a single peak to two peaks is characteristic of the electron pairing transition.** (d) Intensity plot of anti-symmetrized Hall resistance R_{xy}^{anti} versus μ and B . (e) Plot of energy difference between transconductance peaks versus magnetic field. **Blue dashed line extrapolates to a value of $B_p = 3.9 \pm 0.4$ T.** (f) Average R_{xy}^{anti} over the range $\mu = 1.95$ meV to 2.55 meV reveals nonlinear behavior with asymptotes that cross at $B_H = 3.4 \pm 0.5$ T.

We have modified the discussion on electron pairing transition in the main text and the figure captions (as shown above) in the revised manuscript for clarity.

Lines 200-203: “For $|B| < B_p$, transport is governed by a single quantum channel composed of electron pairs that propagate quasi-ballistically. For $|B| > B_p$, the paired channel splits into spin-up and spin-down single-electron channels with subband bottoms that split in energy and appear as two distinct peaks on the transconductance (Figure 2 (c)).”

There are many other prominent features in the data of Fig. 1(b) at the higher value of m . While the derivative plot of Fig. 1(c) is drawn over a very limited range of m , the readers would wonder about the origin of such high energy features, which have been conveniently ignored.

We appreciate that the Referee has shown interest in the higher energy features of our data. In general, when the chemical potential is increased, occupation of multiple subbands takes place, corresponding to lateral and vertical excited modes of the electron waveguide system. This phenomenon is well established. The simplest case to study is the lowest subbands, which is what we focus on. We are not ignoring the other data, but it becomes increasingly difficult to separate the various contributions from subbands at higher chemical potentials.

We have added the transconductance plot over the full range of chemical potential in Figure S2 of the supplementary information.

Figure S2: Nanocross device geometry and longitudinal measurements across nanocross Device A1 oriented at $\varphi = 65^\circ$. (a) Schematic of longitudinal and Hall transport measurements across the nanocross. Angle φ denotes the relative orientation of the nanocross with respect to the crystallographic direction. Longitudinal voltage probes ($V_{L\pm}$) enable four-terminal conductance to be measured while transverse voltage probes ($V_{H\pm}$) enable Hall measurements. Both longitudinal and Hall measurements are acquired simultaneously as a function of gate voltage (V_{sg1} or V_{sg2}) and applied magnetic field, B , (b) Longitudinal conductance G versus chemical potential μ for magnetic fields ranging between $B = 0$ T and $B = 8$ T in steps of 1 T for Device A1 oriented at $\varphi = 65^\circ$ with respect to $[100]$ crystallographic direction. A conductance plateau near $G \approx 1.75 e^2/h$ appears at all magnetic fields. For magnetic fields larger than $B = 4$ T, the transition to this plateau broadens significantly, and a second plateau is clearly visible at $G \approx 0.90 \pm 0.05 e^2/h$ at $B = 8$ T. Curves are offset by $1 e^2/h$ for clarity, (c) Transconductance $dG/d\mu$ versus μ for magnetic fields ranging between $B = 0$ T and $B = 8$ T in steps of 1 T and the full range of chemical potential. $dG/d\mu$ versus μ reveals a transition between paired and unpaired state near $B = 4$ T as shown by the dashed blue lines. Curves are offset for clarity.

Lines 144-146: “The zero-bias longitudinal conductance $G = dI/dV$ (Figure 1(b)) and transconductance $dG/d\mu$ (Figure 1(c) and Figure S2(c)) is shown for Device A1 ($\varphi = 65^\circ$) as a function of μ , for magnetic fields ranging between $B = 0$ T and $B = 8$ T.”

Lines 150-151: “The corresponding line cuts for $dG/d\mu$ (Figure 1(c)) focused on the relevant range of chemical potential shows a clear splitting of the $1.75 e^2/h$ peak starting at $B = 4$ T.”

2. Although the Hall data have a larger spread, one would like to see the angular dependence of the Hall slope and the of critical field at which deviation from linearity occur as a function of angle φ as it is changed from zero to 90 degrees by sculpting several nanochannels.

We appreciate the desire of the Referee to want to see more data. We showed reproducibility of the Hall response for three devices at 65 degrees (Figure 3b), all of which show remarkably similar behavior. We are confident that the behavior at a particular angle is reproducible. It will be nice to look at more angles in follow-up studies, but that data is not necessary in our view to substantiate the claims of the present manuscript.

We have modified the main text and table caption (as shown above) to clarify these details and added Figure S8 showing the angular dependence of the Hall slope R_H and critical field B_H to the supplementary information of the revised manuscript.

Table 1 : Hall transition field, B_H , electron pairing field, B_P , and slope of anomalous Hall response below (R_H^{low}) and above (R_H^{high}) the transition field B_H , summarized for eight nanocross devices A1-E oriented for $0^\circ < \varphi < 90^\circ$.

Nanocross angle φ	Sample	Device	B_P (T)	B_H (T)	R_H^{low} (Ω/T)	R_H^{high} (Ω/T)
65°	1	A1	3.9 ± 0.4	3.4 ± 0.5	43	26
		A2		3.3 ± 0.4	42	24
		A3		3.2 ± 0.3	42	24
0°		B1		5.9 ± 0.1	43	74
		B2		5.2 ± 2.9	44	37
45°		C		1.8 ± 0.2	49	70
75°		D		2.2 ± 0.1	49	76
45°	2	E	2.2 ± 0.4	2.4 ± 0.6	44	32

Lines 251-253: “Additionally, Figure S8 shows the increase in variation of the Hall coefficient, R_H vs φ with increasing magnitude of magnetic field and in-plane anisotropy of critical field B_P vs φ .”

Figure S8: Angle dependence of Hall slope R_H and Hall transition field B_H in LAO/STO at increasing magnetic field strengths. Angle φ denotes the relative orientation of the nanocross with respect to the [100] crystallographic direction. (a) Spline fit showing variation in R_H with increasing magnetic field strength, $1 T \leq B \leq 7 T$ for $0 \leq \varphi \leq 180^\circ$ (b) Spline fit showing variation in critical magnetic field, B_H for $0 \leq \varphi \leq 180^\circ$. The graphs consider two axes of symmetry, rotational symmetry by 90° and mirror symmetry along 45° .

One would also expect the zero and 90-degree result to be degenerate.

We appreciate the reviewer recognizing that zero and 90-degree angles are expected to be degenerate as also highlighted in lines 248-249 of the manuscript.

We have also modified the figure captions of Figure 5 and Figure S8 to highlight this detail.

3. The experimental section does not provide any information about device fabrication. For example, the temperature at which the nanochannels were written is not mentioned.

We thank the reviewer for the comments.

The details about the device fabrication are discussed in the materials and methods section of the manuscript.

We have further modified the materials and methods section to include details regarding the temperature at which the nanochannels are written.

Lines 307-309: “Sixteen interface contacts, formed by milling 25 nm-deep trenches and subsequently depositing Ti/Au (4 nm/25 nm), surround a 25 μm x 25 μm “canvas” where devices are “sketched” with a voltage-biased c-AFM tip at ambient temperature.”

If these structures were sculpted at ambient temperature, then one cannot say what will be their position with respect to the ferroelastic domains as their appearance is a stochastic process occurring at the phase change point.

We thank the reviewer for the comments.

The nanocross devices are created using c-AFM lithography at ambient temperature. Previously, piezoelectric force microscopy (PFM) imaging experiments on LAO/STO nanostructures have shown that conducting paths written by c-AFM lithography are elongated along the Z-axis at room temperature (41). Further, scanning SET measurements on LAO/STO show that while the X and Y domains share a similar surface potential, the Z domains have a higher surface potential, varying by ~ 1 meV (32). Thus, it can be argued that the conductive nanowires created using c-AFM lithography “pre-seed” the formation of Z domains in STO, whereas the X and Y domains define the insulating states.

We have added the above-mentioned details and references regarding the correlation between ferroelastic domains and conductive nanostructures to the revised manuscript.

Lines 107-113: “Lastly, piezoelectric force microscopy imaging of conductive nanowires sketched at the LAO/STO interface using c-AFM lithography reveals that the conducting paths are elongated along the Z-axis at room temperature (41). Furthermore, low temperature scanning SET measurements of LAO/STO show that while the X and Y ferroelastic domains share similar surface potentials, the Z domains have a higher surface potential, varying by approximately 1 meV (32). Thus, it can be argued that the conductive nanowires created using c-AFM lithography “pre-seed” the formation of Z domains in STO, whereas the X and Y domains define the insulating states.”

The literature on ferroelastic domains also suggests that the domain size could be much larger than the size of the device. The domain patterns at the surface of the crystal may also have several orientations (see, for example, J Appl. Phys. 86, 1653 (1999)).

We thank the reviewer for the comments.

The ferroelastic domains discussed in literature refer to naturally occurring ferroelastic domains in STO which can be much larger in domain size and have several orientations. Our study is unique in the sense that we artificially define the X, Y, and Z ferroelastic domains at nanometer length scales by combining a unique conductive AFM lithography procedure with the novel quasi-1D nanocross geometry. The presence of the intersection in the nanocross define both Z-X and Z-Y domain boundaries as shown in Figure 4 of the manuscript. The ability to control and manipulate the X, Y and Z domains at the nanoscale limit allows us to directly study its implication on the unique transport properties at the LAO/STO interface such as the correlation between electron pairing without superconductivity, anomalous Hall effect and electronic nematicity which are inaccessible with conventional macroscopic devices.

We have added modified the text to highlight this novel aspect of the nanocross devices.

Lines 115-128: “Here we describe mesoscopic transport experiments which aim to probe the correlation between electron pairing, AHE, and electronic nematicity in LAO/STO. The LAO/STO samples are grown using pulsed laser deposition details of which are described in the film fabrication section of Materials and Methods. These measurements are enabled by quasi-1D cross-shaped ballistic electron waveguides or “nanocrosses”, created at the LAO/STO interface using c-AFM lithography (20). The nanocross devices serve as a building block to understand 1D electron physics at the LAO/STO interface. The multi-terminal nature of the nanocross allow four-terminal measurements to be performed simultaneously in both longitudinal and Hall configurations, allowing the two distinct physical phenomena to be directly compared (Figure 3 (a)). The unique cross shaped geometry of the nanocross defines both Z–X and Z–Y ferroelastic domain boundaries in the system, in close proximity at the nanoscale limit. We have previously discussed the correlation between nanocross devices and ferroelastic domains (20). Further, the role of angular dependence of electron pairing and AHE is investigated by sculpting nanocross devices at varied angles, ϕ between 0 and 90-degrees with respect to the [100] crystallographic direction (Figure 3 (a)).

4. Supplementary information on the quality of the samples is missing. One would like to know the number of LAO monolayers, concentration, and mobility of charge carriers in the 2D gas, and the propensity of this gas to undergo condensation. Would the sample become superconducting if appropriate gate bias is used?

We thank the reviewer for the comments.

We have added details regarding the quality of the samples: number of LAO monolayers, concentration, and mobility of charge carriers in the 2D gas, to the materials and methods section of the manuscript.

Lines 297-298: “The 3.4-unit cell LAO/STO samples are epitaxially grown on TiO₂-terminated STO (001) substrates using pulsed laser deposition and they are nominally insulating.”

Lines 316-317: “Conductive nanostructures created by c-AFM lithography have a 2D carrier density typically in the range of $0.5\text{--}1.0 \times 10^{13} \text{ cm}^{-2}$ and a 2D electron mobility $\mu_H \sim 10^3 \text{ cm}^2/(\text{V s})$.”

The LAO/STO devices superconducts at temperatures below the superconducting transition temperature $T_c < 300 \text{ mK}$ and at fields below the superconducting critical field $B_c < 0.2 \text{ T}$. However, the critical pairing field B_p of preformed pairs as discussed in this manuscript is much higher than the critical field for superconductivity ($B_p \gg B_c$). Hence, the paired electron states at these high magnetic fields are always non-superconducting irrespective of the gate bias used.

5. Lastly, the classic paper on superconductivity in reduced SrTiO₃ crystals (Ref. # 14) could have been referenced in a better context. It does not say anything about two-dimensional superconductivity or heterointerfaces. This study is on bulk reduced crystals, and in many respects seems to tell us why LAO/STO interface becomes superconducting on high temperature annealing and on bombardment by energetic species ejected from the target during growth.

We thank the reviewer for pointing out the issue with Ref 14 (Ref 13 in revised manuscript).

We have amended the revised manuscript and cited the reference in a more appropriate context.

Lines 47-49: “Strontium titanate (STO) is the first and best-known superconducting semiconductor (13). STO-based heterostructures, and in particular formed with LaAlO₃ (LAO) (14) inherits the superconducting properties from STO and exhibit two-dimensional (2D) superconductivity without the need for chemical doping (15).

Reviewer #2 (Remarks to the Author):

This manuscript by the group of Jeremy Levy reports on a study of magneto transport in crossbar nanodevices of LaAlO₃/SrTiO₃ interface 2DEGs. The main finding of this work is a correlation between the onset of magnetic field-dependent anomalies in the Hall effect and the onset of strong angle dependence in the magnetoresistance which shows structure indicative of a Fermi surface created from underlying highly anisotropic orbitals.

Their results suggest a pair-breaking induced density reorganization between d_{xy} and d_{xz}/d_{yz} orbitals as a function of the magnetic field.

The term ‘nematicity’ is not correctly applied here since in a crystal this terminology must refer to a state where some underlying crystal symmetry is spontaneously broken. Principally, the

presence of angular variations in the field dependent AMR does not imply nematic order which is what the authors seem to be claiming since it is not clear that the crystalline C4 symmetry is being broken to C2.

We thank the reviewer for raising this point. The term "nematicity" commonly refers to when liquid crystals spontaneously align under an electric field leading to a crystalline symmetry breaking. Electron nematicity unlike conventional nematicity is characterized by the rotational symmetry breaking of an electronic fluid due to anisotropic electron interactions resulting in strongly anisotropic transport behavior which can be tunable with chemical potential or chemical doping, and also by a magnetic field. It has been detected experimentally in several materials, including two-dimensional electron gas (3,4), high Tc superconductors (5-7), and twisted bilayer graphene (8) by probing various electronic properties. Under this definition, we believe that our study meets the criteria for electronic nematicity, in that the in-plane anisotropy appears above a critical magnetic field and does not precisely coincide with the crystallographic directions. We argue that the emergence of in-plane anisotropy at the LAO/STO interface is due to the transition of electrons from the d_{xy} paired state to the d_{xz} and d_{yz} unpaired states, with the latter exhibiting a high degree of anisotropic behavior.

We have modified the text to clarify this important concept of electron nematicity.

Lines 33-40: "The term nematicity was first used in the context to liquid crystals to describe the system's crystalline symmetry breaking from C4 to C2. Over time the definition of nematicity has evolved to other domains including the electronic nematic phases based on various theoretical models (1). Electronic nematicity is characterized by the rotational symmetry breaking of an electronic fluid due to anisotropic electron interactions, resulting in strongly anisotropic transport behavior which can be tunable with chemical potential or chemical doping, and also by a magnetic field (2). Electronic nematic phases have been found to exist in a wide range of electronic materials (1) extending from GaAs/AlGaAs heterostructures (3, 4) to high-temperature superconductors (5-7) and twisted bilayer graphene (8)."

The second issue is that much of the information upon which this claim is based is not new.

We agree with the Referee that our claims are based on established information obtained from research in this field by our group and by other groups. Our claims, which rest on this established foundation, are new.

(i) There has been an old observation of anomaly in the AMR by the Ilani group back in 2013 (ref.24) at B~3T where crystalline components become visible. This was identified as being possibly due to new orbitals becoming important - speculated but not substantiated as being via a lattice Kondo effect.

We agree about the importance of the work published by the Ilani group in identifying an emergence of magnetization at the LAO/STO interface presumably due to Kondo effect, and have cited it where appropriate. However, the source of magnetization responsible for the effects has remained unidentified. Our experimental findings point to a specific origin of excess magnetization, one which is associated with the breaking of spin-singlet electron pairs. Above the pairing field, spin-singlet electron pairs unbind and spin-polarize, resulting in characteristic changes in the Hall response. The results reported here provide direct evidence in support of this mechanism, associating the AHE with the pairing transition. We further study the angle-dependence of the anomalous Hall response which reveals an onset of electronic nematicity that again coincides with the electron pairing transition, unveiling a rotational symmetry breaking due to the transition from paired to unpaired phases at the interface. The results presented here highlights the influence of preformed electron pairs on the transport properties of LAO/STO in mediating electronic nematicity in STO-based systems.

(ii) Several previous works by J. Levy group has already shown that there are conductance plateaus in LAO/STO nanostructures whose details (conductance values, plateau widths, etc) depend on the participating sub-bands and device geometry.

Indeed, the interpretation of results presented here are based on our understanding of electron waveguides. Without the Hall cross geometry, it is not possible to relate the ballistic transport of electron pairs (and their unbinding above a pairing field B_p) to the anomalous Hall effect. Additionally, the presence of the intersection in the nanocross devices defines both Z-X and Z-Y ferroelastic domain boundaries in the system, in close proximity thereby providing more stringent boundary conditions to study the role of ferroelastic domains.

To help emphasize the above points, we have modified the summary paragraph highlighting the novelty of the results and physics discovered.

Lines 119-128: *“The nanocross devices serve as a building block to understand 1D electron physics at the LAO/STO interface. The multi-terminal nature of the nanocross allow four-terminal measurements to be performed simultaneously in both longitudinal and Hall configurations, allowing the two distinct physical phenomena to be directly compared (Figure 1(a)). The cross shaped geometry of the nanocross defines both Z-X and Z-Y ferroelastic domain boundaries in the system, in close proximity at the nanoscale limit. Further, the role of angular dependence of electron pairing and AHE is investigated by sculpting nanocross devices at varied angles, φ between 0 and 90-degrees with respect to the [100] crystallographic direction (Figure 1(a)).”*

Lines 273-292: *“In summary, simultaneous longitudinal and Hall measurements on novel quasi-1D ballistic nanocrosses sketched at the LAO/STO interface show a direct correlation between the electron pairing transition and nonlinearities in the Hall response. The correlation between electron pairing and AHE is independent of the orientation of the nanocross with respect to the*

crystallographic direction and of the magnitude of electron pairing field. Angle-dependent Hall measurements taken across multiple nanocross orientations further show evidence of electronic nematicity whose onset also coincides with the pairing transition. A natural explanation is connecting the electron pairing transition to a shift between d_{xy} electron pairs and d_{xz} and d_{yz} unpaired states, with the latter exhibiting a high degree of anisotropic behavior. The correlation between electron pairing, AHE and electronic nematicity consolidates a wide range of seemingly disparate experimental findings reported in STO and construct a comprehensive understanding of the rich correlated nanoelectronics present in this system. The results presented in this work provide several new insights regarding this system; the significant role of ferroelastic domains as elusive pairing “glue” in STO and the importance of the 1D paired liquid phase, in general the pre-formed pairs on the transport properties in LAO/STO. Although the existence of the paired liquid states at the LAO/STO interface has been known for half a decade now, they still fail to find a place in the phase space of STO-based heterostructures. The given results reinforce the need to go beyond the single-particle descriptions and consider these pre-formed pairs as an essential element of the phase diagram of STO-based systems. These results can possibly be extended to other correlated systems and non-conventional superconductors where pre-formed pairs are known to exist but not considered while studying the transport phenomena.”

(iii) The main observation here seems to be Fig.1c where a splitting of a central peak at $\mu \sim 2\text{meV}$ is seen for $B \sim 3\text{T}$. However, taken by itself, it is not clear that this implies that it is a pairing gap (presumably from d_{xy} orbital based on the manuscript) evolving into Zeeman split gap as opposed to, say, a Kondo hybridization gap.

We thank the reviewer for the comments. The interpretation of this central peak splitting is based on dozens of splitting's observed in straight electron waveguide structures and described in multiple papers. We do not believe that the introduction of a nanocross geometry will introduce a new physical feature that mimics the one identified for straight electron waveguides. Regarding the Kondo interpretation, it was considered in Ref. (18). While the Kondo ridge like Zeeman splitting can occur above a critical magnetic field these splitting's are generally observed at non-zero biases. Furthermore, other Kondo parity signatures were not observed here ruling out the Kondo hybridization gap theory.

In addition, orbital dependent peaks are not seen in this figure. If additional orbitals are becoming populated or depopulated at this field, why are no additional features seen associated with them around these energies?

We thank the reviewer for the comments. The transconductance spectra represents the subband structure of the system. A peak in the transconductance marks the chemical potential at which a new subband contributes to transport. The subbands are further separated by regions where the conductance is quantized ($dG/d\mu \sim 0$). The blue dashed lines in Figure 1c shows the splitting of

the subbands thereby representing the magnetic depopulation effect. This is also represented in Figure 2c where the splitting of the peaks is overlaid with the transconductance spectra.

Furthermore, the data is shown for a single device and not for various devices and at various temperatures to show that this field scale is correlated in AMR and in these tunneling cross-conductance.

We appreciate this comment, which was also raised by the other referee. Here we reproduce our response:

The electron nematicity at the LAO/STO interface is represented over seven devices (A1-A3, B1-B2, C, D) for multiple angles $\varphi = 0^\circ, 45^\circ, 65^\circ$, and 75° . The critical magnetic field, B_H is also shown to be highly reproducible for nominally identical devices written at the same location and orientation on the sample during different conductive atomic force microscope lithography cycles. Further, the correlation between electron pairing and anomalous Hall effect is shown on two samples for two different orientations of the nanocross device, $f = 65^\circ$ and 45° and pairing field B_p differing by a factor of 2 ($B_p = 3.4 \pm 0.4 T$ and $2.2 \pm 0.4 T$). The correlation between

electron pairing, anomalous Hall effect and electron nematicity shown over a large set devices with significantly different pairing field and multiple values of angle φ is strong evidence to support our central claim.

Concerning the question about the temperature dependence, such an investigation would not be possible since quantized ballistic transport, one of the key quantities being measured, is not measurable at significantly elevated temperatures. The electronic subbands gets blurred at higher temperatures and hence one cannot resolve the pairing field, B_p .

Lines 126-128: “Further, the role of angular dependence of electron pairing and AHE is investigated by sculpting nanocross devices at varied angles, φ between 0 and 90-degrees with respect to the [100] crystallographic direction (Figure 1(a)).”

Lines 173-177: “Figure 3 shows Hall measurements performed as a function of the orientation of the nanocross with respect to the [100] crystallographic direction (denoted by angle φ in Figure 3(a)) taken across seven nanocross devices. We investigate two important aspects here: (a) the reproducibility of Hall measurements at a given location and orientation of the nanocross on the sample, and (b) the angular dependence of the Hall response.”

Lines 210-213: “The striking agreement between the pairing field and anomalous Hall response in LAO/STO heterostructures across nanocross devices sculpted on two different samples at different orientations with respect to the [100] crystallographic direction and pairing field differing by a factor of two suggest an underlying physical mechanism that relates them.”

Lines 239-242: “The results presented here across nanocross devices at varied angles with respect to the crystallographic direction give further empirical evidence linking the preformed electron pairs, AHE and ferroelastic domain structures in LAO/STO.”

Lines 275-279: “The correlation between electron pairing and AHE is independent of the orientation of the nanocross with respect to the crystallographic direction and of the magnitude of electron pairing field. Angle-dependent Hall measurements taken across multiple nanocross orientations further show evidence of electronic nematicity whose onset also coincides with the pairing transition.”

Overall, I think the paper makes an interesting but still very speculative suggestion. If there was extensive new data or new physics being discovered, it might have been more reasonable to consider this for Nature Communications.

We believe that we have provided strong evidence for a direct correlation between two very different transport experiments. This correlation connects the electron pairing transition with anomalous Hall effect. These two effects have never been probed simultaneously in nanoscale

devices before. The experiments were painstaking, because multiple devices had to be created in precisely the same location, all while changing the angle of the Hall cross. Our findings show that there is a connection, and we provide a reasonable physical picture to describe the behavior. Our experimental findings have important implications for understanding the nature of correlated electrons in the LAO/STO system.

Reviewer #3 (Remarks to the Author):

The study conducted by Aditi Nethewala and colleagues explores the transport properties of quasi-1D structures on SrTiO₃/LaAlO₃ interfaces fabricated using a conducting atomic force microscope. The authors establish correlations between conductance versus gate voltage, Hall signal, and direction dependence of the Hall effect to uncover the underlying transport mechanisms.

The authors interpret the correlation between the field at which the Hall becomes anisotropic and the vanishing of features in the conductance characteristics as evidence for the importance of preformed pairs in the transport properties of STO-Based heterostructures. The authors claim that a single energy scale governs the transport features of these interfaces.

While some of the features have been reported, for example, Figure 1 shows data similar to previous publications from the same group (e.g., Ref 17), the new merit of this contribution is the correlation between the features.

While the findings are noteworthy, some concerns remain. These issues require clarification before publication in Nature Communications.

We thank the reviewer for the positive feedback over the importance of our manuscript.

A potential issue with the argument put forth by Nethewala et al. is the apparent temperature dependence of various transport features, as seen in previous publications and cited within this study. Specifically, the anomalous Hall and anisotropic magnetoresistance have been observed to persist up to approximately 30 K, which is a characteristic temperature associated with the formation of ferroelastic polar domains in SrTiO₃. It is, therefore, unclear whether the observed pairs that condense at 300 mK could form at temperatures two orders of magnitude higher. It would be valuable for the authors to demonstrate whether the depairing field, nematicity, and anomalous Hall exhibit the same temperature dependence to clarify this matter.

We thank the reviewer for the comments.

In the given manuscript we have presented two important results:

1. Evidence revealing that the onset of electronic nematicity in the LAO/STO-based nanostructures coincides with the electron pairing transition, unveiling a rotational symmetry breaking due to the transition from paired to unpaired phases at the interface.

2. A ferroelastic domain model that shows good agreement with our experimental observations and reveals that the observed minima and maxima of the Hall transition field, B_H coincides with the two extreme ferroelastic domain configurations across the nanocross.

However, a temperature dependence investigation would not be possible in the given case since quantized ballistic transport, one of the key quantities being measured, is not measurable at significantly elevated temperatures. The electronic subbands gets blurred at higher temperatures and hence one cannot resolve the pairing field, B_p .

I do not understand why the Hall slope in figure S2 does not change with gate voltage. Doesn't that mean that the side gates are ineffective in changing the carrier density?

We thank the reviewer for the comments.

The change in sidegate voltage in case of nanocross devices is less than a volt. Hence, we do not expect the sidegate voltage to significantly alter the carrier density of the system like a 2D Hall bar where the corresponding change in backgate voltage is at the order of several hundreds of volts. Additionally, any small change in the carrier density of the quasi-1D nanocross devices considered here will also be dominated by scattering effects and hence difficult to distinguish. We do observe the conductance of the nanocross to tune from 0 to $\sim 4 e^2/h$ (Figure 1b) which represents the change in chemical potential of the system due to the applied side gate voltage.

Due to the nonlinear dielectric constant, the chemical potential, μ , is not necessarily proportional to the gate voltage. The lever arm model is, I believe, relevant for the linear response, which does not apply to STO

We appreciate the concern of the referees concerning the validity of the lever arm model. This concern would be appropriate if we were using the back gate to scan the chemical potential of our device. Instead, the nanocross devices are tuned by applying a side gate voltage not backgate voltage. A constant small ($\sim 1V$) back gate voltage is maintained to tune the device overall, but it is held fixed while the side gate is tuned. The lever arm is defined with respect to the side gate, not the back gate. Hence, the lever arm model works well for this class of nanocross devices which are tuned by side gates.

In figure 5, it is not clear what the fit is based on, there are only a few data points, if I understand correctly, yet the fit is very detailed. Can the authors show the data the fit is based on?

We thank the reviewer for the comments.

Figure 5 has been updated to show the data points the fit is based on. The graph assumes two axes of symmetry, rotational symmetry by 90° and mirror symmetry along 45° , and is interpolated between measured values.

Figure 5: Angle dependence of Hall response and electron nematicity in LAO/STO at increasing magnetic field strengths. Angle φ denotes the relative orientation of the nanocross with respect to the [100] crystallographic direction. (a) Spline fit overlaid with the experimental data points show the variation in R_{xy}^{anti} with increasing magnetic field strength, $1 T \leq B \leq 7 T$ for $0 \leq \varphi \leq 180^\circ$. The graph considers two axes of symmetry, rotational symmetry by 90° and mirror symmetry along 45° , (b) Evolution of nematicity marker $N(B) = \langle \Delta R_{xy}^{anti^2} \rangle^{1/2}$ as a function of magnitude of magnetic field B .

After the authors have thoroughly addressed the aforementioned concerns, I will be in a position to recommend the publication of their research in Nature Communications.

REVIEWERS' COMMENTS

Reviewer #2 (Remarks to the Author):

The authors appear to have addressed all the points raised in my report and seem to have also addressed questions raised by the other referees. I'm ok with accepting this for publication in Nat Com.

Reviewer #3 (Remarks to the Author):

The authors have properly addressed the comments of the reviewers. They explained why they cannot study the temperature dependence and this limits their possibility to further prove their claims.

I am still wandering whether the pre-formed pairs scenario is the simplest one to explain the data.

The paper will stimulate further discussion and experiments and I suggest publication in Nature Communications.